# Mercury and Prenatal Growth: A Systematic Review

**DOI:** 10.3390/ijerph18137140

**Published:** 2021-07-03

**Authors:** Kyle Dack, Matthew Fell, Caroline M. Taylor, Alexandra Havdahl, Sarah J. Lewis

**Affiliations:** 1Medical Research Council Integrative Epidemiology Unit, University of Bristol, Bristol BS8 2BN, UK; s.j.lewis@bristol.ac.uk; 2Cleft Collective, University of Bristol, Bristol BS8 2BN, UK; mattfell@doctors.org.uk; 3Centre for Academic Child Health, Bristol Medical School, University of Bristol, Bristol BS8 1NU, UK; Caroline.m.taylor@bristol.ac.uk; 4Department of Mental Disorders, Norwegian Institute of Public Health, 0456 Oslo, Norway; Alexandra.havdahl@fhi.no; 5Nic Waals Institute, Lovisenberg Diaconal Hospital, 0771 Oslo, Norway; 6Population Health Sciences, Bristol Medical School, University of Bristol, Bristol BS8 2BN, UK

**Keywords:** systematic review, pregnancy, childhood, mercury, toxic metal, birth weight, birth length, head circumference, prenatal growth

## Abstract

The intrauterine environment is critical for healthy prenatal growth and affects neonatal survival and later health. Mercury is a toxic metal which can freely cross the placenta and disrupt a wide range of cellular processes. Many observational studies have investigated mercury exposure and prenatal growth, but no prior review has synthesised this evidence. Four relevant publication databases (Embase, MEDLINE/PubMed, PsycINFO, and Scopus) were systematically searched to identify studies of prenatal mercury exposure and birth weight, birth length, or head circumference. Study quality was assessed using the NIH Quality Assessment Tool, and results synthesised in a narrative review. Twenty-seven studies met the review criteria, these were in 17 countries and used 8 types of mercury biomarker. Studies of birth weight (total = 27) involving populations with high levels of mercury exposure, non-linear methods, or identified as high quality were more likely to report an association with mercury, but overall results were inconsistent. Most studies reported no strong evidence of association between mercury and birth length (n = 14) or head circumference (n = 14). Overall, our review did not identify strong evidence that mercury exposure leads to impaired prenatal growth, although there was some evidence of a negative association of mercury with birth weight.

## 1. Introduction

Prenatal development involves the growth and development of foetal internal systems and is implicated in a range of later health outcomes. Overall prenatal growth and intrauterine health are commonly benchmarked after birth using neonatal birth weight, birth length (crown to heel), head circumference, and other anthropometric measures [1]. The prevalence of births classed as low birth weight (<2500 g) globally in 2015 was 14.6% [2] and the rate of decline is not on course to meet WHO targets [3]. Growth measures at birth are predictive of infant mortality [4], risk of childhood illness and impaired development [5], and may even be associated with long-term adult health [6]. The most up to date (2015) report on childhood mortality estimated that 45% of deaths (2.6 million) below the age of 5 years were neonates, most linked to prenatal or intrapartum conditions [7]. 

Prenatal development is dependent upon an optimal intrauterine environment and foetal responses to this environment, which are mediated by mechanisms such as the production of foetal and placental hormones, modified blood flow, and metabolic changes [8]. The developmental environment is also linked to maternal reproductive health and is sensitive to maternal nutrition and contact with harmful environmental agents [8]. Identifying hazardous substances and reducing exposure to them may be an effective way of improving overall neonatal health and survival [9]. 

The toxic metal mercury (Hg) is one such substance of concern and is frequently flagged in antenatal care guidance usually in the context of advice on fish consumption [10]. Mercury is abundant in the earth’s crust and is mobilised into the environment through human industrial activity and natural events such as volcanic activity [11,12,13]. Routes to human contact are well documented [13,14,15], and include the food chain, soil, water contamination, and mercury-containing cosmetics and other goods [16]. 

The human toxicity of mercury varies depending on the duration of exposure, dosage, and the specific compound of mercury. Elemental mercury when ingested is relatively benign due to low gastrointestinal absorption, but when inhaled can enter the blood stream where it can cross both the placenta and blood–brain barrier. In contrast, inorganic mercury when ingested tends to accumulate within the kidneys and can also be formed within the body from oxidised elemental mercury. Both forms are highly reactive with sulphur-containing proteins and can deactivate enzymes, inhibit DNA methylation and cell division, and lead to oxidative stress and cell death [14,17]. Methylmercury (MeHg)—an organic compound bound with carbon—is the most bioavailable form of mercury and can easily enter the body through the digestion of contaminated foods [14]. MeHg is lipid soluble and can be distributed throughout the body with a high reactivity with sulfhydryl groups, leading to disruption of a wide range of basic cellular functions [18]. 

Mercury may be a threat to the developing foetus because both elemental and organic forms of mercury can cross the placenta during gestation [11,18], where it may accumulate in a far higher dose-to-weight ratio than is possible in an adult. The removal and excretion of mercury is relatively slow: in adults elemental mercury has a biological half-life of 35–90 days, inorganic mercury approximately 40 days, and methylmercury approximately 65 days [18]. Therefore, prolonged exposure may lead to the accumulation of harmful quantities. However, whether low level mercury exposure during pregnancy harms foetal growth to any measurable degree is unclear. There have been many observational studies investigating this question, but no prior meta-analyses or systematic reviews have been published. 

Governmental and non-governmental organisation guidelines for fish consumption as a source of mercury exposure vary, in terms of quantity and species to avoid [10]. This is in part due to uncertainty about the cost/benefits of associated nutrients and elements in fish, including mercury. Much of the focus of previous research has been on the neurodevelopmental impact of mercury exposure. However, the widespread reactivity of mercury makes it possible that exposure interferes with overall foetal growth and not only the brain. The aim of this study is to review the evidence on mercury exposure during pregnancy and prenatal growth. Specific objectives are: (1) to systematically search for studies of prenatal mercury exposure and perinatal growth outcomes; (2) to evaluate the quality of the existing evidence; (3) to synthesise the evidence and compare for differences based on quality and other methodological dimensions. 

## 2. Materials and Methods

### 2.1. Protocol and Study Design

In brief, we aimed to identify prospective and cross-sectional studies of birth outcomes, which measured biomarkers of mercury exposure in either the mother or neonate. Our criteria allow for studies which used a cross-sectional design measuring mercury in biomarkers at birth, because current tissue concentrations reflect past exposure, and there do not appear to be biological mechanisms which make reverse causation a plausible possibility. The design and methodology of this systematic review were registered with the International Prospective Register of Systematic Reviews (PROSPERO) on 23rd November 2020, registration number CRD42020221146. Deviations from the protocol are listed in Appendix A. 

### 2.2. Search Strategy

We searched Embase, MEDLINE (PubMed), PsycINFO, and Scopus article databases on 01 December 2020. We identified indexed subject headings (where available) and keywords related to: (1)Pregnancy and early childhood;(2)Mercury;(3)Foetal growth.

Animal studies were excluded using filters, and there was no restriction on year of publication or language. Search methods varied depending on the functionality of each database, and the terms and searches used are provided in Appendix A. We checked reference lists of included papers for further articles. We additionally used Google search to identify unpublished literature such as white papers, Ph.D. dissertations or theses, or conference proceedings, where the full paper was digitally available. Including unpublished literature has the advantage of identifying studies where publication bias is less likely. Grey literature was subject to the same inclusion criteria and quality assessment as published studies. 

### 2.3. Study Selection

Duplicates were automatically identified and results removed using Covidence systematic review software (www.covidence.org, Accessed on 1 January 2020). The remaining results were screened by two reviewers (KD and MF) by title and abstract against the criteria in Table 1. Papers potentially meeting the inclusion criteria were read in full before a final inclusion/exclusion decision was taken. Where the two reviewers disagreed, they discussed the paper to reach a consensus agreement. The discordance rate in both the first and second stage of screening was approximately 10%.

### 2.4. Data Extraction and Quality Assessment

The following data were extracted from the included studies: study author, location, mercury measurement method, study design, population characteristics, types of birth outcomes, statistical methods, and study results (full list in Appendix A). There were some studies which overall had a prospective design but were categorised as cross-sectional because the exposure and outcome of interest for this review were measured at the same time., i.e., if mercury was measured at delivery. The extracted data were checked by the second reviewer. 

Study quality was evaluated using the National Institutes of Health (NIH) Quality Assessment Tool [19], which is valid for cross-sectional and cohort studies. This contains 14 yes/no questions which evaluate the sampling, design, and reporting quality of studies (Appendix A). Two additional criteria were added to evaluate common analytical or reporting practices which could encourage bias, making 16 criteria in total. The first was designed to record whether studies selected covariates based on prior theory or significance testing, which may inadequately control for confounding [20]. The second recorded whether studies reported all results and not only those which met a certain level of statistical significance. Studies which (a) met 12 or more of the NIH quality criteria and (b) met criterion 14—“Were key potential confounding variables measured and adjusted statistically for their impact on the relationship between exposure (s) and out-come (s)?”—were classified as “high quality”. Twelve was chosen as the cut-off a priori as a way of identifying those studies which were of higher quality, because there is no standard cut-off. Criterion 14 was considered essential because studies which did not adjust for key confounders are likely to have biased results that exaggerate or obscure any association.

### 2.5. Evidence Synthesis

It was not possible to meta-analyse results due to heterogeneity in the mercury measurements and model parameterisation. Instead, we created a narrative synthesis based on results from the identified studies. 

To supplement our review, we used albatross plots to visualise results from studies based on their direction of effect and *p*-value [21]. These plots are intended to aid the readers’ understanding of the approximate pattern of results and should be considered a visual display of the data rather than replacement for a meta-analysis. Separate plots were created for birth weight, birth length, and head circumference. 

Where studies had results from multiple mercury measurements (e.g., hair and whole blood), each biomarker is represented once per plot and colour-coded so readers can compare the same measure between studies and identify similarities or differences between different measures. When a study did not report a *p*-value, confidence intervals were used to calculate one [22]. If a study did not report the exact *p*-value and it was not possible to calculate it, for the purpose of the albatross plot *p* was assigned as the reported threshold minus 0.001. For example, Al-Saleh et al. (2014) reported one estimate with a significance of *p* < 0.05 which could not be plotted without a specific *p*-value, so we used *p* = 0.05 − 0.001 = 0.049. All confidence intervals are reported using the 95% lower and upper intervals. 

## 3. Results

### 3.1. Summary

A flowchart of the search and screening process is shown in Figure 1. Our searches identified 193 potentially eligible studies, of which 27 met the inclusion criteria. Study characteristics are summarised in Table 2. Our 27 included studies were conducted in 17 different countries, most (16 studies) used a cross-sectional design and the remainder (11 studies) were prospective. All 27 studies evaluated mercury exposure and birth weight, and approximately half (14 studies) also estimated associations with birth length and head circumference. A variety of mercury measurement methods were used (see Appendix A for details), and the lower limit of mercury detection (LoD) varied considerably from a highly sensitive 0.00004 μg/g MeHg in maternal hair [23] to 2 μg/L Hg in cord blood [24,25]. Full details of each study are reported in Appendix B.

Most studies measured mercury concentrations in umbilical cord blood or tissue (13 studies), whole blood (9 studies) and maternal hair (6 studies) taken either during antenatal care visits or at the time of delivery. Results in this review are grouped by biological matrix reflecting representation of a different pathway and duration of exposure. If foetal growth is more sensitive to mercury exposure in a particular period of development, this could lead to a systematic difference in results between different biomarkers. Details of these differences are in Appendix C. 

Each section of the following review is structured to take these differences into account for each outcome: (1)Summary of all results.(2)Results from circulating blood (whole, serum, or erythrocyte) and urine mercury concentrations.(3)Results from umbilical cord blood or tissue and neonatal hair samples.(4)Results from placental tissue and maternal hair samples taken at birth.(5)Results compared by study quality and study design, and an overall conclusion.

### 3.2. Review

#### 3.2.1. Birth Weight

Twenty-seven studies estimated the association between mercury and birth weight. All primary results were extracted from these studies (Appendix A). Many studies reported more than one primary effect estimate: a total of 49 results were reported across the 27 studies. A total of 33 estimates from 22 studies reported no evidence of an association [23,24,25,26,27,28,29,30,31,32,33,34,35,36,37,38,39,40,41,42,43,44]. Evidence of an inverse association between mercury and birth weight was found in 16 estimates from 11 studies [25,34,36,37,40,41,45,46,47,48,49] and possible patterns between studies that may explain these findings will be explored in the following paragraphs. Figure 2 plots the number of participants, direction of effect, and *p*-value of 36 of these results, with each study represented once per biomarker (details in Appendix A). While most of the plotted results clustered around the null, there is a substantial minority of results on the left side of the graph which reflect an inverse association between mercury and birth weight. No study reported strong evidence of a positive association between mercury and birth weight. 

The first group of results included investigations of mercury concentrations using whole blood or urine during the first trimester [28,35,37,41,49]. Three prospective studies of pregnancies in South Korea [37], fishing areas of the USA [28], and Los Angeles, USA [35] each found no evidence of an association with birthweight. A UK-based cohort study reported similar results from mothers who consumed fish, but a negative association in mothers with diets lacking fish (−57 g per μg Hg/L, CI: −112.5 to −1.5) [41]. Evidence of an inverse association between maternal whole blood mercury in the first trimester and birth weight was also reported from a Japanese prospective study (−0.17 S.D units per Log μg/L, *p* = 0.006) [49]. The women in this study reported eating fish 5 times per week and had higher mean mercury concentrations (6.06 μg/L) than other studies of first trimester whole blood (range: 1.02 to 2.07 μg/L). 

Whole blood and urine were also measured in the third trimester or at delivery [27,29,30,34,36,37,46,47,48]. Four of these studies reported no strong evidence of an association with birth weight [27,29,30,37], but results elsewhere were mixed. Three cross-sectional studies of whole blood taken at delivery in China, Greenland, and Nigeria [34,46,47] each reported negative correlations between mercury content and birth weight, as did a Swedish cohort using erythrocyte mercury [48]. 

One possible reason for these different results is that the studies involved populations with different degrees of mercury exposure. The central tendency measures of mercury exposure from each study differ considerably (Appendix A). However, for studies using blood samples in the third trimester or at delivery, populations with similar mean or median mercury exposure still reported different findings. Two studies with mean Hg values of 0.91 μg/L and 3.3 μg/L reported no evidence of an association with birthweight (0.48 g per Log10 μg/L, CI: −303 to 304; −65.5 g per Log μg/L, CI: −135.5 to 4.5) [29,37]. However, a doctoral dissertation which studied rural pregnancies in China with a mean Hg concentration of 1.5 μg/L reported evidence of an inverse relationship with birth weight (−0.36 SD units per Log10 μg/L, CI: −0.73 to 0.01) [34]. Two further studies identified possible non-linear relationships; a Swedish cohort with a median Hg of 1.5 μg/kg (High Hg: −59 g per μg/kg, CI: −115 to −3; Low Hg: 58 g, 11 to 105) [48] and a South Korean/Taiwanese cohort with a median Hg of 3.27 μg/L (>25th percentile:−58 g per doubling Hg, CI: −100 to −10) [36] in both studies an inverse association was found only in the “High Hg” group. Finally, a cohort based in Greenland found a mean Hg concentration of 14.9 μg/L which is much higher than any other in this group of studies, owing to the local diet being heavily dependent on marine food [47]. This study adjusted for key confounders including seafood consumption, and birth weight was −7.1 g lower per μg/L of blood mercury (*p* = 0.019). 

The second group of studies to be reviewed used newborn hair or umbilical cord blood/tissue samples [24,25,27,29,32,33,36,37,38,39,42,43,45,47]. Nine reported no strong evidence of an association with birth weight [24,27,29,32,33,38,39,42,43]. This included a cohort of 182 births in the Faroe Islands with high seafood consumption [32] and high mean umbilical cord blood concentration (101.7 nmol/L, equivalent to 20.44 μg/L). No effect size was reported from this study, only a *p*-value of 0.63. Several studies reported evidence of a negative association with birth weight, including the previously discussed Greenland study [47] which also measured umbilical cord (−4.2 g per μg/L, *p* = 0.012). Inverse associations were also reported in a prospective South Korean cohort (−86.4 g per Log μg/L, CI: −163.1 to 9.7) [37] and cross-sectional study in Saudi Arabia (−0.03 SD units per μg/L, *p* < 0.05) [45], although the former may have been biased by not adjusting for fish consumption. Non-linearity was explored in two studies by dividing the pregnancy cohort into quartiles of umbilical cord mercury concentrations [25,36]. The first, using two cohorts in South Korea and Taiwan, reported that a doubling of mercury in the top quartile was associated with a change in birth weight of −77 g (CI: −125 to −29). The second, a Spanish cohort which met our criteria for high study quality, reported negative associations in the top 3 Hg quartiles compared to the lowest quartile (4th: −100 g, CI: −200 to −0.5; 3rd: −76.7g, CI: −179.2 to 25.8; 2nd: −100.3g, −200.2 to −0.5) [25]. 

Finally, seven studies used maternal hair samples or placental tissue [23,26,31,33,34,38,44], biomarkers which may be most representative of mercury exposure across the whole pregnancy. Four studies using maternal hair [23,26,38,44] and two using placental tissue (one using both) [31,37] all reported no strong evidence of an association between mercury and birth weight. Two of these studies were conducted in Suriname and the Amazon basin, where the authors speculated that Hg exposure may have occurred, respectively, from fresh-water fishing near small scale gold mining [26] and tin mining activities [38]. Studies in the Seychelles and coastal China reported that the expected route of exposure was diets high in ocean fish [33,44]. One study of rural Chinese women reported that hair mercury concentrations were associated with birth weight (−0.41 SD units per Log10 μg/g, CI: −0.78 to −0.032), the evidence for an effect based on hair methylmercury was not as strong, but suggestive of an effect in the same direction (−0.31 SD units per Log10 μg/g, CI: −0.63 to 0.0) [34]. This study scored highly in our quality assessment and adjusted for key confounders. 

Each study was assessed using the NIH Quality Assessment Tool, with scores ranging from 8–16 out of a maximum of 16 (scoring in Appendix A). Mercury concentrations were measured using biomarkers and laboratory scientists measuring the concentrations were unlikely to be aware of birth outcomes, further reducing any small possibility of observer bias. The most common differences between studies were in the clarity of inclusion criteria (item 4), explanation of sample size (item 5), the reporting of quality controls in the mercury sample analysis (item 10), and the adjustment of key confounders (item 14). Our review of prior literature determined that among the many factors with a potential role in foetal growth, three were the most likely to be confounding factors. These were maternal socio-economic status or education, fish or fatty acid intake, and maternal smoking status, each of which is expected to influence both prenatal mercury exposure and foetal growth [50,51,52,53]. Studies which met our high quality criteria (n = 10) are plotted in Figure 3. There appears to be slightly more evidence for a negative association between mercury and birth weight when considering only these studies. 

Overall, studies mostly reported no strong evidence of an association between prenatal mercury exposure and birth weight. However, we found that high quality studies were more likely to report evidence of a negative association. Additional methodological factors also appeared to be important. Studies that explored non-linearity or had populations with relatively higher mean levels of mercury more frequently found a negative association, which may suggest a threshold effect. Additionally, even among studies that did not report strong evidence of an association, most estimates were negative. As seen in Appendix A, very few studies reported a positive estimate between mercury and birth weight, and none reported strong evidence of such a result. Smaller studies were more likely to report evidence of a negative association, so we assessed the possibility of publication bias in results where we had sufficient studies using the biological matrix and outcome using Egger regression tests. We found no strong evidence that publication was related to the direction of effect for any outcome (data in Appendix A). 

#### 3.2.2. Birth Length

Fourteen studies investigated birth length and prenatal mercury exposure. A total of 10 studies reported no evidence of an association [24,25,27,29,33,34,39,41,43,44], 2 found evidence of both positive and negative associations in different matrices [28,45], and 2 reported evidence of a negative association [31,48]. This can be seen in Figure 4.

Whole blood or erythrocytes were used in five studies [28,29,34,41,48] and urine in one [27]. The only study using erythrocytes reported evidence of a positive association in women with Hg < 1.0 μg/kg (0.24 cm per μg/kg, CI: 0.023 to 0.4), and a negative association in women with Hg > 1.0 μg/kg (−0.29 cm, CI: −0.54 to −0.05) [48]. A US cohort used tertiles of pre-pregnancy mercury to identify evidence of a positive dose response association [28]. In both studies, the authors suggested this may be indicative of fish consumption and that Hg levels may have acted as a proxy for long-chain polyunsaturated fatty acids (LCPUFA) exposure which had a positive effect on growth. However, two studies of populations which consumed high quantities of fish in New York [27] and Laizhou Bay, China [29], which did not adjust for fish consumption in their models did not find evidence of a positive association with birth length although in both cases confidence intervals were wide (New York: −1.74 cm per 10% increase in Hg, CI: −6.2 to 2.71; China: 0.39 cm per Log10 μg/L, −0.85 to 1.63). A UK cohort which stratified between fish eaters and non-fish eaters [41] and was assessed high quality according to our criteria reported that neither group’s mercury concentrations showed evidence of an association with birth length. Similarly, a second study from China [34] reported no association after adjustment for fish consumption (0.04 SD units per Log10 μg/L, CI: −0.44 to 0.51). Overall, most studies using these biological matrices do not report evidence of an association between mercury and birth length. 

All studies of mercury in neonatal hair [33] and umbilical cord blood or tissue samples [24,25,27,29,33,39,43,45] reported no strong evidence of an association with birth length. Mercury concentrations reported in these studies had a large range and included two Spanish cohorts [24,25] with comparatively high geometric mean concentrations (8.2 and 9.4 μg/L). The pattern of results was not different when looking at only high quality studies. Five high quality studies reported no strong evidence of an association [24,34,43,44,45], and two reported both evidence of a negative association and other results that indicated no association [25,45]. 

Three studies used maternal hair samples, none of which reported strong evidence of an association with birth length [33,34,44]. These studies were located in coastal China (−0.34 cm per μg/kg, CI: −0.76 to 0.46) [33], rural inland China (Hg: −0.28 SD units per Log10 μg/g, CI: −0.72 to 0.15; MeHg: −0.2, 0.6 to 0.21) [34], and the Republic of Seychelles (−0.02 SD units per μg/g, CI: −0.072 to 0.031) [44]. Results from placental tissue were less consistent [31,33,45]. Two studies reported a negative association with Hg per dry placenta weight (−0.22 SD units per μg/g, *p* < 0.01) [45], and births which were born with mercury above the limit of detection in the placenta (−0.47 cm, CI: −0.99 to 0.05) [31]. However, a third study with relatively similar mean concentrations of placental mercury reported almost no change in birth length (0.04 cm per μg/kg, CI: −0.51 to 0.93) [33]. 

The pattern of results was similar when looking at only high quality studies (n = 6), with the overwhelming majority reporting no strong evidence of an association (Data and figure in Appendix A). In summary most of the above studies report no association between mercury and birth length, and this is consistent across mercury biomarkers, statistical methods, study quality, and adjustment for key confounding variables. Although two studies reported evidence of a non-linear negative association [28,48], these findings failed to be replicated elsewhere.

#### 3.2.3. Head Circumference

Fourteen studies of prenatal mercury exposure and head circumference were found. A visualisation of these results in the albatross plot Figure 5 shows that almost all results appear to indicate no strong evidence of an association. The complete results are listed in Appendix A: 1 study reported mixed evidence depending on the biomarker used [34], and 13 others reported evidence of no association with head circumference [24,27,28,29,30,31,33,39,41,43,44,45,48]. 

Mercury in blood samples taken pre-pregnancy [28], in the first trimester [41], and late pregnancy [29,30,48] were not found to be associated with head circumference at birth. Additionally, a US birth cohort reported an estimated association close to zero from third trimester urine mercury concentrations (3.63 cm per 10% increase in Hg, CI: −66.8 to 74.1) [27]. In a UK cohort sub-group analysis investigated fish eaters (0.0 cm per μg/L, CI: −0.05 to 0.05) and non-fish eaters (−0.05 cm per μg/L, CI: −0.24 to 0.15), neither of which showed strong evidence of an association with whole blood mercury [41]. A single study reported evidence of a negative correlation between whole blood mercury and head circumference SD units, with each Log10 μg/L increase in mercury being associated with −0.13 SD units (CI: −0.24 to −0.02) [34]. The population lived in an area of low mercury exposure (mean 1.5 μg/L) which does not deviate considerably from other studies which measured blood mercury (Appendix A for comparisons). 

No other measure of mercury exposure had strong evidence of an association with head circumference. This includes one analysis of neonatal hair (−0.12 cm per μg/kg, CI: −0.82 to 0.04) [33], seven of umbilical cord samples [24,27,29,33,39,41,45], three of maternal hair [33,34,44], and two of placental tissue samples [31,33]. Of the five studies we considered to be high quality, one reported both strong and weak evidence depending on biological matrix [34], and four no evidence of an association [24,39,41,43]. There are too few studies to draw strong conclusions from this, but most estimates were negative. Overall, studies in this review indicated that prenatal mercury exposure is not associated with head circumference, but evidence from high quality studies was mixed. The lack of a relationship remained consistent despite significant heterogeneity in study location, method, mean mercury concentration, and quality. 

## 4. Discussion

This review identified a diverse range of studies including populations including populations together encompassing a wide range of Hg concentrations. Biomarkers were used that correspond with mercury exposure during preconception, early, late, and the full term of pregnancy.

When considering only studies that adjusted for key confounders and were considered high quality, a greater proportion of results reported strong evidence of a negative association of mercury concentrations with birthweight. Overall, most model estimates were negative, and a minority of analyses found stronger evidence of a negative association. However, taken together most analyses from these studies did not indicate strong evidence of an association between maternal mercury concentrations and birth weight. Confidence intervals which did not overlap with the null were reported in 16 estimates from 11 studies, out of a total of 49 estimates from 27 studies. These analyses tended to use subgroups by Hg levels or use cohorts with relatively higher mean mercury concentrations than other studies of the same tissue, but this pattern was inconsistent. 

A clearer pattern was seen with birth length and head circumference. Almost all studies reported no strong evidence that prenatal mercury exposure is associated with birth length (10 of 14 studies), or head circumference (13 of 14 studies). While there were a few studies of these two outcomes that reported a negative association, it may be that unidentified confounding or other methodological challenges have influenced their results. The lack of strong evidence was similar when considering only high quality studies, although studies tended to show negative associations which overlapped with the null particularly results from high quality studies of head circumference. It is therefore not possible to rule out a small negative effect which studies were underpowered to detect. 

The prevalence of mercury in the environment and well publicised dangers from contamination events has led to chronic mercury exposure at lower levels being a concern for both health professionals and pregnant women. Both NHS (National Health Service, in England) and CDC (Centers for Disease Control and Prevention, in the USA) guidelines, among others, identify mercury as a danger during pregnancy and recommend avoiding excessive consumption of mercury-containing foods [54,55]. This review does not find strong evidence that lower levels of exposure common to most countries is associated with prenatal growth. If this is accurate, it is possible that guidance to avoid dietary mercury may be unnecessary or overly complicated for any marginal gain in birth outcomes. Such guidance is not without potential harms, including the loss of the many beneficial nutrients contained in fish [56].

Two major events have informed much of our awareness of acute mercury toxicity: seed grain contamination in Basra, Iraq, and coastal pollution in Minamata, Japan [13]. Both resulted in detrimental effects on foetal growth and infant mortality. However, in both cases the exposure was qualitatively different from that of chronic exposure from diet. In Basra, grain stores were treated with a fungicide containing organic mercury species, which led to mass poisoning in the following months [57]. In Minamata, an industrial plant contaminated the bay and marine life within it with organic mercury compounds [58]. In contrast, chronic dietary exposure as studied in this review frequently involves bio-methylation of elemental or inorganic species into methylmercury. Secondly, both contamination events involved far higher levels of mercury accumulation than any study in this review reported: hair mercury concentrations peaked in Basra at 120 to 600 μg/g [59] and in Minamata a maximum of 705 μg/g was reported [60]. It may be that the harms seen from acute poisoning are not generalisable to chronic exposure. 

The strengths of this study are firstly that it is the first systematic review of the large number of studies which were conducted on prenatal mercury exposure and foetal growth. Secondly, a wide variety of populations were involved and study authors reported the predominant modes of exposure to include cooking with liquefied natural gas [46], rice consumption [34], local fish consumption in areas contaminated with small scale gold or tin mining [26,38], and seafood consumption ([e.g., [47]). It also includes populations with Western diets that have relatively low levels of mercury exposure (e.g., [23,28,41]). Thirdly, we were able to identify studies which used a variety of types of tissue to measure prenatal mercury exposure. These can contain different forms of mercury and are sensitive to different durations of mercury exposure. Finally, we followed methods pre-specified by a protocol and used tools such as albatross plots to review the evidence in a systematic manner. By doing so, we have avoided focusing solely on the positive or most interesting results from each study. 

There are several possible limitations to this review. First, the heterogeneity of biomarkers used to measure mercury made it difficult to compare results between all studies. We identified two comprehensive reviews of mercury biomarkers which informed our comparisons [61,62], but there is a degree of uncertainty in our current knowledge of how mercury accumulates and is distributed across tissues. Consequently, it was not possible to compare mean mercury concentrations between biomarkers, and we were unable to identify with confidence which study populations experienced the highest levels of prenatal mercury exposure. This impaired our ability to make judgements of potential non-linear effects at higher levels of mercury exposure. 

Secondly, we used albatross plots to visually represent study findings, but some studies reported more than one estimate for a single biomarker and birth outcome. These could not be plotted together, and criteria were developed to select one estimate per study. Given the range of differences between studies, these criteria may have led to bias in the plots. For this reason, the plots should be considered a visual aid to the review rather than a complete visual representation of the evidence. A third limitation is that because of the large number of results, we limited this review to only continuous measures of birth outcomes. Binary outcomes such as low birth weight were excluded, and it is possible that the evidence may differ in studies of these. Finally, this review could not assess possible interactions between mercury and other toxicants or nutrients. Toxic metals can have a synergistic effect [63], and micronutrients such as selenium can protect against them [64]. In practice very few studies measure multiple metals, and in those included in this review [31,39] interaction effects were not statistically tested. 

## 5. Conclusions

From the numerous cross-sectional and prospective studies of mercury and foetal growth, many show no strong evidence of an effect, but a significant minority report inverse associations with birth weight, particularly studies of populations with the highest mean mercury concentrations. Gaps remain in our understanding of the interpretation of different mercury biomarkers and of possible interaction effects. 

## Figures and Tables

**Figure 1 ijerph-18-07140-f001:**
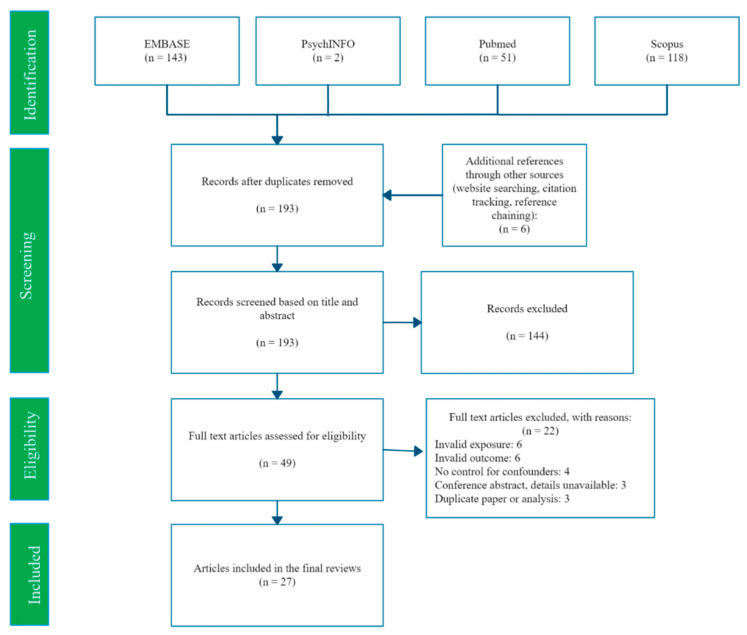
PRISMA flow diagram of search and selection process.

**Figure 2 ijerph-18-07140-f002:**
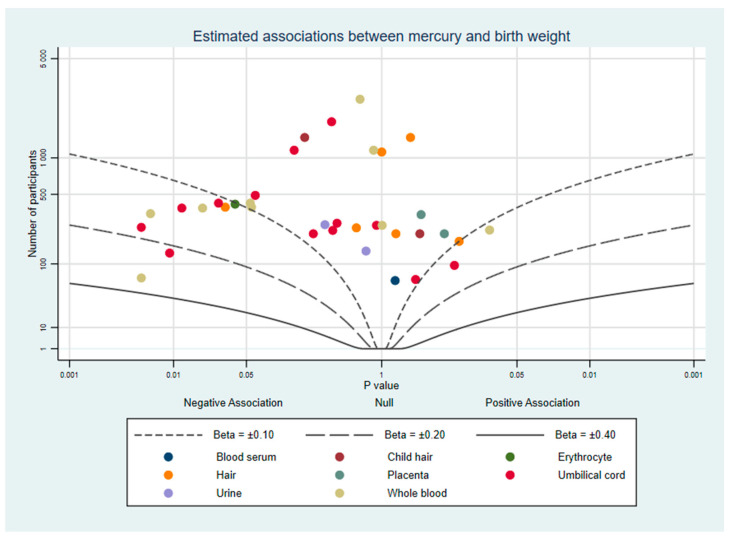
Albatross plot of the estimated effect of mercury exposure on birth weight.

**Figure 3 ijerph-18-07140-f003:**
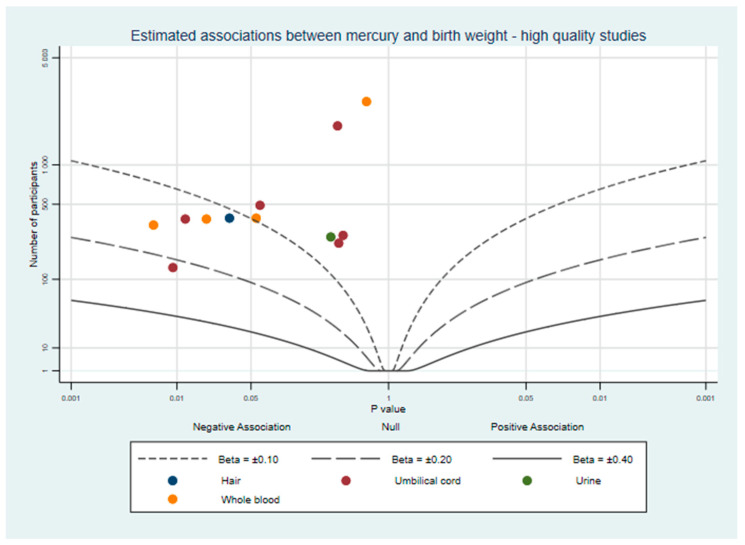
Albatross plot of the estimated effect of mercury exposure on birth weight, from studies which adjusted for maternal socio-economic status or education, fish or fatty acid intake, and maternal smoking status.

**Figure 4 ijerph-18-07140-f004:**
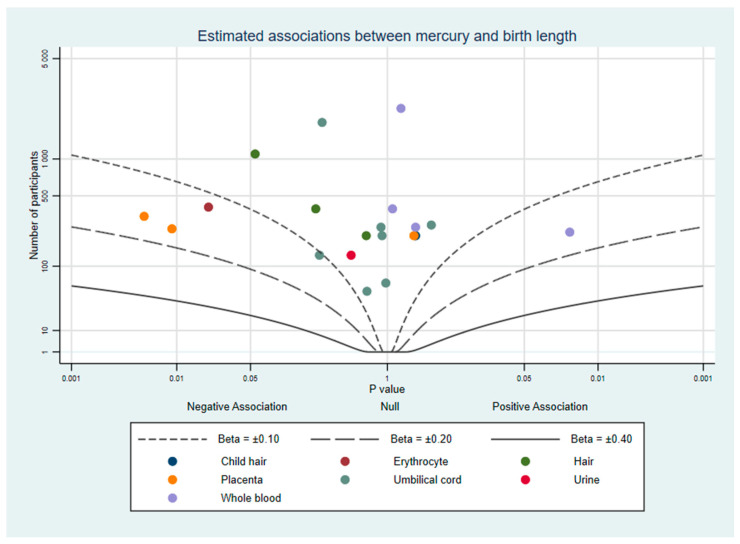
Albatross plot of the estimated effect of mercury exposure on birth length.

**Figure 5 ijerph-18-07140-f005:**
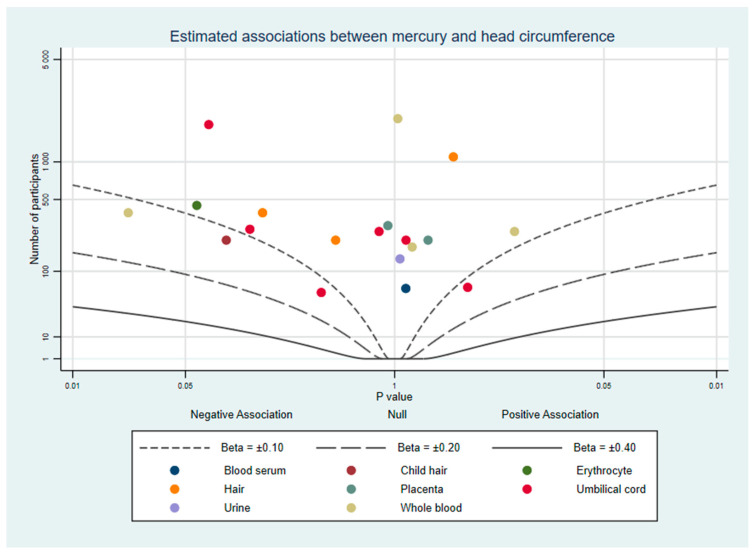
Albatross plot of the estimated effect of mercury exposure on head circumference.

**Table 1 ijerph-18-07140-t001:** Criteria for including or excluding papers from this systematic review.

Include	Exclude
1. Study of total mercury, inorganic, organic, and/or methylmercury compounds.	1. Studies other compounds including ethylmercury.
2. Measures mercury in pregnant women, new-born infants.	2. Measures mercury in other populations.
3. Measures mercury concentrations in biological matrices: blood (whole, erythrocyte, plasma, serum), urine, cord blood/tissue, placenta, and/or hair.	3. Uses any other measure of mercury exposure.
5. Reports association between mercury and either birthweight, birth length, and/or birth head circumference.	5. Does not report associations between mercury and specified outcomes.
6. Study reports results from multivariable analysis methods.	6. Study reports results only from univariable methods such as correlations or *t*-tests.
7. Study of humans.	7. Animal or cellular study.

**Table 2 ijerph-18-07140-t002:** Study characteristics (Studies may be counted in multiple categories).

**Study Design**	***n***	**Outcome**	***n***
Cross-sectional	16	Birth weight	27
Prospective	11	Birth length	14
		Head circumference	14
**Country**	***n***		***n***
Belgium	1	Saudi Arabia	1
Brazil	1	South Korea	2
China	3	Spain	3
Faroe Islands	1	Suriname	1
Greenland	1	Sweden	1
Jamaica	1	Taiwan	1
Japan	3	United Kingdom	1
Nigeria	1	USA	4
Republic of Seychelles	2		
**Sampled Matrix**	***n***	**Mercury Analysis Method**	***n***
Maternal whole blood	9	Atomic absorption spectroscopy (AAS)	10
Maternal blood serum	1	Cold vapor atomic absorption spectrometry (CVAAS)	6
Maternal erythrocyte	1	Cold vapor atomic fluorescence spectroscopy (CV-AFS)	1
Maternal hair	6	Headspace gas chromatography atomic fluorescence spectrometry (HG-GC-AFS)	1
Maternal urine	2	Inductively coupled plasma mass spectrometry (ICP-MS)	9
Umbilical cord (blood or tissue)	13	Not stated	1
Placenta	3		
Child hair	2		

## Data Availability

All extracted data used to write this review are included in Appendix B and Appendix A.

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
