# Peer review of "Mercury and Prenatal Growth: A Systematic Review"

_ijerph, 2021, doi:10.3390/ijerph18137140_

Round 1

Reviewer 1 Report

Reviewer comments on Dack et al.:

Article: Mercury and prenatal growth: a systematic review

The aim of this manuscript is to analyze the linkage between prenatal exposure to Mercury and its impact on fetal growth.

Please find below a list of comments on my review of the manuscript:

  1. Introduction

“Prenatal development is dependent on optimal nutritional and environmental conditions, and identifying modifiable environmental factors may be an effective way of improving overall neonatal health and survival.”

The manuscript may benefit from highlighting that fetal development is deeply linked to mother’s reproductive health. In this context, also female reproductive health is very sensitive to environmental hazards,  such as heavy metals, persistent organic (Exposure to persistent organic pollutants during tooth formation: molecular mechanisms and clinical findings – 2020) and endocrine disruptor compounds, which exert long-term effects on the female reproductive competence and on prenatal growth (Association between female reproductive health and mancozeb: Systematic review of experimental models – 2020).

MATERIAL AND METHODS:

As regards this section, the methodology design was appropriately implemented within the study.

RESULTS:

This section is well organized and densely presented, based on well-synthetized data.

DISCUSSION:

Also this section is well designed and provide recent and complete evidence on the topic.

To sum up, the topic is timely and call for attention. Overall, the manuscript requires minor changes (as mentioned). I would accept the manuscript, if the comments are addressed properly.

Author Response

Thank you for your consideration of our manuscript, and suggestions for improvement. We have made adjustments to take into account your feedback as well as general edits to improve the quality of the manuscript. Edits specific to your feedback are listed below. 

Kind regards,

Kyle Dack

Reviewer 1:

1.1.  The manuscript may benefit from highlighting that fetal development is deeply linked to mother’s reproductive health. In this context, also female reproductive health is very sensitive to environmental hazards,  such as heavy metals, persistent organic (Exposure to persistent organic pollutants during tooth formation: molecular mechanisms and clinical findings – 2020) and endocrine disruptor compounds, which exert long-term effects on the female reproductive competence and on prenatal growth (Association between female reproductive health and mancozeb: Systematic review of experimental models – 2020).

We thank the reviewer for this comment and have modified introduction paragraph 2 to expand on relationship between fetal development, intrauterine environment, and environmental exposures.

“Prenatal development is dependent upon an optimal intrauterine environment and fetal responses to this environment, which are mediated by mechanisms such as the pro-duction of fetal and placental hormones, modified blood flow, and metabolic changes [8]. The developmental environment is also linked to maternal reproductive health and is sensitive to maternal nutrition and contact with harmful environmental agents [8]. Identifying hazardous substances and reducing exposure to them may be an effective way of improving overall neonatal health and survival [9].”

Reviewer 2 Report

The authors systematically review the association of Hg exposure with prenatal growth.  The study is comprehensive and the reader can easily follow their reasoning and conclusions. It is an elegant study

However, there is one paragraph that detracts from an otherwise excellent manuscript and which the authors should reconsider. On page 2 the author’s state:

“Mercury is abundant in the earth’s crust and is mobilized into the environment through human industrial activity and natural events such as volcanic activity [10-12]. Routes to human contact are well documented [12-14], and include the food chain, soil, water contamination, and mercury-containing cosmetics and goods [15]. Once inside the body, mercury is highly reactive disrupting basic cellular functions. Elemental and inorganic mercury can deactivate proteins and enzymes, and inhibit DNA methylation and cell division, leading to oxidative stress and cell death [13, 16]. Methylmercury (MeHg), which we are primarily exposed to through the consumption of large fish, has similar toxic mechanisms and can disrupt a wide range of basic cellular functions [17]. This is a possible threat to the developing fetus because most forms of mercury can cross the placenta during gestation and before metabolic pathways that protect against mercury toxicity are fully developed [10].”

Starting with “Once inside the body…” a bias appears.  As stated, Hg is in the earth’s crust and we all are exposed. However, there is no evidence that “once inside the body” anything happens.  Indeed it would be strange if evolution overlooked something adverse and relied on epidemiologists to find it.  All of the other statements are true, but omit important toxicological points. The toxicity of Hg like other toxicants varies with the specific species, dosage, age at exposure, and duration of exposure. Mercury is a theoretical threat from fish consumption that is as yet unproven.  The classic case of poisoning from fish consumption, Minamata disease turns out to have been misinterpreted or perhaps misrepresented.  Fish at Minamata averaged 10 ppm and had Hg levels to over 100 ppm (Yokoyama 2018).  In addition, the exposure was not from environmentally converted MeHg, but from a mixture of industrially produced organic Hg species (James 2019). It bears no resemblance to consumption of ocean fish with naturally acquired MeHg. In my opinion reference 10 presents a personal biased opinion and I am not aware of any scientific evidence to support the statement "...before metabolic pathways against mercury toxicity are fully developed".  

Reference 17 is incorrect, the author is Broussard and FACB is part of his credentials. I did not check the others, but suggest the authors do so. 

Apart from this paragraph and the reference, this is a very scientific and well done manuscript.  Kudos to the authors.  

Author Response

Thank you for your consideration of our manuscript, and suggestions for improvement. We have made adjustments to take into account your feedback as well as general edits to improve the quality of the manuscript. Edits specific to your feedback are listed below. 

Kind regards,

Kyle Dack

Reviewer 2:

2.1. However, there is one paragraph that detracts from an otherwise excellent manuscript and which the authors should reconsider. On page 2 the author’s state: “..”.

Starting with “Once inside the body…” a bias appears.  As stated, Hg is in the earth’s crust and we all are exposed. However, there is no evidence that “once inside the body” anything happens.  Indeed it would be strange if evolution overlooked something adverse and relied on epidemiologists to find it. 

This statement has been removed.

2.2. All of the other statements are true, but omit important toxicological points. The toxicity of Hg like other toxicants varies with the specific species, dosage, age at exposure, and duration of exposure. Mercury is a theoretical threat from fish consumption that is as yet unproven. 

We agree with the reviewer that more nuance and details of mercury exposure would enhance the manuscript. Introduction paragraph 4 has been expanded with a greater focus on the pathway from exposure to potential fetal harm, including more details on differences between mercury species.

“The human toxicity of mercury varies depending on the duration of exposure, dosage, and the specific compound of mercury. Elemental mercury when ingested is relatively benign due to low gastrointestinal absorption, but when inhaled can enter the blood stream where it can cross both the placenta and blood-brain barrier. In contrast, inorganic mercury when ingested tends to accumulate within the kidneys, and can also be formed within the body from oxidised elemental mercury. Both forms are highly re-active with sulphur-containing proteins and can deactivate enzymes, inhibit DNA methylation and cell division, and lead to oxidative stress and cell death [14,17]. Methylmercury (MeHg) – an organic compound bound with carbon - is the most bioavailable form of mercury and can easily enter the body through the digestion of contaminated foods [14]. MeHg is lipid soluble, and can be distributed throughout the body with a high reactivity with sulfhydryl groups, leading to disruption of a wide range of basic cellular functions [18].”

2.3. The classic case of poisoning from fish consumption, Minamata disease turns out to have been misinterpreted or perhaps misrepresented.  Fish at Minamata averaged 10 ppm and had Hg levels to over 100 ppm (Yokoyama 2018).  In addition, the exposure was not from environmentally converted MeHg, but from a mixture of industrially produced organic Hg species (James 2019). It bears no resemblance to consumption of ocean fish with naturally acquired MeHg.

In response to this feedback and comments from other reviewers, discussion paragraph 5 has been modified to highlight how both Minamata and Basra events are notably different from chronic mercury exposure through dietary sources.

“Two major events have informed much of our awareness of acute mercury toxicity: seed grain contamination in Basra, Iraq, and coastal pollution in Minamata, Japan [13]. Both resulted in detrimental effects on fetal growth and infant mortality. However, in both cases the exposure was qualitatively different from that of chronic exposure from diet. In Basra, grain stores were treated with a fungicide containing organic mercury species, which led to mass poisoning in the following months [57]. In Minamata, an industrial plant contaminated the bay and marine life within it with organic mercury compounds [58]. In contrast, chronic dietary exposure as studied in this review frequently involves bio-methylation of elemental or inorganic species into methylmercury. Secondly, both contamination events involved far higher levels of mercury accumulation than any study in this review reported: hair mercury concentrations peaked in Basra at 120 to 600 μg/g [59] and in Minamata a maximum of 705 μg/g was reported [60]. It may be that the harms seen from acute poisoning are not generalizable to chronic exposure.”

2.4. In my opinion reference 10 presents a personal biased opinion and I am not aware of any scientific evidence to support the statement "...before metabolic pathways against mercury toxicity are fully developed".  

This statement has been removed and introduction paragraph 5 rewritten with an alternative explanation for fetal mercury toxicity.

“Mercury may be a threat to the developing fetus because both elemental and organic forms of mercury can cross the placenta during gestation [11,18], where it may accumulate in a far higher dose-to-weight ratio than is possible in an adult. The removal and excretion of mercury is relatively slow: in adults elemental mercury has a biological half-life of 35-90 days, inorganic mercury approximately 40 days, and methylmercury approximately 65 days [18]. Therefore prolonged exposure may lead to the accumulation of harmful quantities.”

2.5. Reference 17 is incorrect, the author is Broussard and FACB is part of his credentials. I did not check the others, but suggest the authors do so. 

Thank you, this has been fixed and all other references checked.

Reviewer 3 Report

Title: Mercury and prenatal growth: a systematic review. Dack et al.

Objective: To review the evidence on mercury exposure during pregnancy and prenatal growth. Specific objectives are to (1) systematically search for studies of prenatal mercury exposure and perinatal growth outcomes; (2) evaluate the quality of the existing evidence; (3) synthesize the evidence and compare for differences based on quality and other methodological dimensions.

General comments:

The topic is per se interesting and it tackles the existing evidence addressing maternal environmental Hg exposure and prenatal growth. The narrative is well written but some assumptions are overlooked or not properly addressed, especially related to “quality and other methodological dimensions”. Thus, it has the potential to be reference for many health issues associated with environmental Hg and birth outcome. My main concern is that pathways of maternal Hg exposure likely to chronically influence fetal growth are not part of the “quality and other methodological dimensions”.

MeHg, the Hg chemical species that is highly absorbed and retained, and it is likely to be found in fish/seafood and rice. Therefore, some aspects of addressing these sources of environmental exposure in relation to the interesting findings are sorely missed throughout the manuscript. This cannot be overlooked: interactions leading to positive, negative, neutralizing effects will impact on results/interpretation.

Specific comments:

The specific foods (fish and rice) can be both carriers of essential nutrients and pollutants (EDC), thus capable of influencing (positively/negatively) fetal growth. Potential benefits and risks associated with maternal consumption of these foods need to be appropriately contextualized.

Here are two easily found references that might help clarify the raised issues: <https://pubmed.ncbi.nlm.nih.gov/18653214/> < https://pubmed.ncbi.nlm.nih.gov/31522043/>

These are only examples of easily found references; the authors may find better suited ones.

Page 5: “The type of biological sample is important because each differs with respect to mercury accumulated, metabolism, and half-life…” Better to use ‘matrix’ instead of sample. Half-life is only mentioned here but in no place it is discussed. There should be a thorough discussion of what these analyzed matrices (blood, hair, etc…) represent in relation to measured birth outcomes (pathway and time of exposure). Hair-Hg (mostly MeHg) represents chronic exposure while blood-Hg represents recent (acute) exposure; they represent different body Hg half-lives that measured during pregnancy or perinatally can tell different stories. This cannot be ignored.

Pg 7: When discussing Hg in matrices, when/if information is available, it is important to inform the dietary origin of MeHg in Western and Asian studies (fish or rice respectively).

Pg 7: Belgium, Surinam, and Seychelles are countries with nuanced sources of Hg exposures: Belgian mothers probably buy commercial fish/seafood, Surinam mothers could be subsistence/traditional villagers (fresh water small fish), and Seychelles mothers probably consume Ocean fish.

Pg 8: Publication bias: did you look at pathways (fish/seafood and rice)?

Pg 11: The events of Iraq and Minamata are valid concerns but they represent quite different scenarios:

-Iraq: Mass poisoning caused by intentionally added fungicide (ethyl- and methyl-Hg) to wheat seeds that were inadvertently/accidentally consumed.

-Minamata: Accumulation of chloro-alkali plant effluents (that contained organic-Hg compounds) in seafood heavily consumed by villagers in post-war Japan with a rice-based diet.

The authors somewhat raised these concerns, but not sufficiently to ward off concerns for acute/chronic Hg exposure in modern-day urban consumers.

Pg 12: “…and areas near industrial mercury use (e.g., [31,43])”. This sentence needs attention/clarification. The “industrial Hg use” refers to artisanal small-scale gold mining (ASGM) activities and loosely refers to fish in certain areas to have high-Hg concentrations for the case of Surinam [31] and in the case of [43] no attribution of fish-Hg consumption is related to ASGM activities. Please rewrite the sentence more accurately after reading the cited references.

Table 2: Instead of “mercury source” use ‘sampled matrix’; mercury source conveys the idea of exposure pathways which should be explored. Please fix “exposure” in Appendix A.

Reference 39: Hong (2017) is a Doctoral dissertation referenced without year of publication. Because it is not a peer reviewed article (like the other studies) it should be justified or better, deleted.

Author Response

Thank you for your consideration of our manuscript, and suggestions for improvement. We have made adjustments to take into account your feedback as well as general edits to improve the quality of the manuscript. Edits specific to your feedback are listed below. 

Kind regards,

Kyle Dack

Reviewer 3:

3.1. “The type of biological sample is important because each differs with respect to mercury accumulated, metabolism, and half-life…” Better to use ‘matrix’ instead of sample.

The phrasing has been changed as suggested.

3.2. Half-life is only mentioned here but in no place it is discussed. There should be a thorough discussion of what these analyzed matrices (blood, hair, etc…) represent in relation to measured birth outcomes (pathway and time of exposure). Hair-Hg (mostly MeHg) represents chronic exposure while blood-Hg represents recent (acute) exposure; they represent different body Hg half-lives that measured during pregnancy or perinatally can tell different stories. This cannot be ignored. There should be a thorough discussion of what these analyzed matrices (blood, hair, etc…) represent in relation to measured birth outcomes (pathway and time of exposure). Hair-Hg (mostly MeHg) represents chronic exposure while blood-Hg represents recent (acute) exposure; they represent different body Hg half-lives that measured during pregnancy or perinatally can tell different stories. This cannot be ignored.

We have added to introduction paragraphs 4 & 5 an expanded discussion of the pathways and biological half-lives of Hg matrices.

“The human toxicity of mercury varies depending on the duration of exposure, dosage, and the specific compound of mercury. Elemental mercury when ingested is relatively benign due to low gastrointestinal absorption, but when inhaled can enter the blood stream where it can cross both the placenta and blood-brain barrier. In contrast, inorganic mercury when ingested tends to accumulate within the kidneys, and can also be formed within the body from oxidised elemental mercury. Both forms are highly re-active with sulphur-containing proteins and can deactivate enzymes, inhibit DNA methylation and cell division, and lead to oxidative stress and cell death [14,17]. Methylmercury (MeHg) – an organic compound bound with carbon - is the most bioavailable form of mercury and can easily enter the body through the digestion of contaminated foods [14]. MeHg is lipid soluble, and can be distributed throughout the body with a high reactivity with sulfhydryl groups, leading to disruption of a wide range of basic cellular functions [18].

Mercury may be a threat to the developing fetus because both elemental and organic forms of mercury can cross the placenta during gestation [11,18], where it may accumulate in a far higher dose-to-weight ratio than is possible in an adult. The removal and excretion of mercury is relatively slow: in adults elemental mercury has a biological half-life of 35-90 days, inorganic mercury approximately 40 days, and methylmercury approximately 65 days [18].”

A detailed description of differences between mercury matrices was added in Appendix B, referenced at the beginning of the results. This discusses the pathways, duration of exposure, and relevance of each biological matrix.

Upon absorption, mercury is rapidly transported in blood throughout the body. Blood mercury concentrations are therefore sensitive to short term changes in exposure [65], although levels may remain elevated for weeks after exposure [66]. Most methylmercury in the blood is bound to erythrocytes, and in pregnancy maternal blood mercury is known to be correlated with umbilical cord blood levels [61]. Urinary mercury has utility in measuring mid to long term elemental or inorganic mercury exposure, but sensitivity to methylmercury exposure is poor because of low levels of excretion [61,65].

The umbilical cord tends to primarily accumulate methylmercury, which during gestation can cross the placenta to the fetus [67] It is estimated that upon delivery, cord blood or tissue concentrations reflect mercury exposure across the third trimester [67]. Neonatal hair which begins growing in the third trimester may also be used to identify methylmercury exposure during this time, because hair contains amino acids which readily and permanently bind with MeHg [65].

Maternal hair which has grown throughout the pregnancy reflects longer-term methylmercury exposure [65], and it tends to correlate well with maternal and fetal blood Hg [61,62]. Concentrations of Hg in the placenta correlate with fetal Hg, and this matrix is most useful to measure elemental and methylmercury which can both cross the placental barrier [68]. Few studies have looked at placental mercury accumulation, but there are some indications that it may accumulate throughout the duration of the pregnancy [68,69].”

3.3. Belgium, Surinam, and Seychelles are countries with nuanced sources of Hg exposures: Belgian mothers probably buy commercial fish/seafood, Surinam mothers could be subsistence/traditional villagers (fresh water small fish), and Seychelles mothers probably consume Ocean fish.

The path of mercury exposure is an interesting factor and could play a role in explaining differences between the reviewed studies. However, whilst a few studies have reported data on correlations between sample Hg and prior factors such as gas cooking or fish intake, most studies only speculate on mercury sources in their introduction or discussion sections.

We have addressed this comment in 2 ways:

  1. We have added a discussion on possible sources of exposure to the specific paragraph on page 8 paragraph 3, but overall we did not feel that these were identified in a reliable enough way for this to be added throughout the results and review.

“Two of these studies were conducted in Suriname and the Amazon basin, where the authors speculated that Hg exposure may have occurred respectively from fresh water fishing near small scale gold mining [26] and tin mining activities [38]. Studies in the Seychelles and coastal China reported that the expected route of exposure was diets high in ocean fish [33,44]. One study of rural Chinese women reported that hair mercury concentrations were associated with birth weight (-0.41 SD units per Log10 μg/g, CI: -0.78 to -0.032), the evidence for an effect based on hair methylmercury was not as strong, but suggestive of an effect in the same direction (-0.31 SD units per Log10 μg/g, CI: -0.63 to 0.0) [34]. This study scored highly in our quality assessment and adjusted for key confounders. “ 

  1. Modified the quality assessment criteria so that “high quality” studies were defined as those which score at least 12 on our quality assessment tool and which also adjusted for key confounders, including fish consumption which appears to be the primary possible pathway for most studies. We think this is a more reliable way to identify which studies could have biased estimates. The findings from this were slightly stronger evidence of a negative association with birth weight, and head circumference which is now reflect in our overall conclusions.

This change is present throughout the paper.

3.4. Pg 8: Publication bias: did you look at pathways (fish/seafood and rice)?

In addition to the above, an Egger regression test of small study bias has been added where there were enough studies for a biological matrix & outcome combination. Exposure pathways could not be added to this analysis of publication bias because while the majority of studies speculated on the source of mercury (the vast majority identifying seafood or river fish), in most cases this was not supported by any data.

"Smaller studies were more likely to report evidence of a negative association, so we assessed the possibility of publication bias in results where we had sufficient studies using the biological matrix and outcome using Egger regression tests. We found no strong evidence that publication was related to the direction of effect for any outcome (data in Supplementary File Part 10)."

3.5. Pg 11: The events of Iraq and Minamata are valid concerns but they represent quite different scenarios:

-Iraq: Mass poisoning caused by intentionally added fungicide (ethyl- and methyl-Hg) to wheat seeds that were inadvertently/accidentally consumed.

-Minamata: Accumulation of chloro-alkali plant effluents (that contained organic-Hg compounds) in seafood heavily consumed by villagers in post-war Japan with a rice-based diet.

The authors somewhat raised these concerns, but not sufficiently to ward off concerns for acute/chronic Hg exposure in modern-day urban consumers.

In response to this feedback and comments from other reviewers, paragraphs 3&4 in the discussion have been modified to highlight how both Minamata and Basra events are notably different from chronic mercury exposure through dietary sources.

 “Two major events have informed much of our awareness of acute mercury toxicity: seed grain contamination in Basra, Iraq, and coastal pollution in Minamata, Japan [13]. Both resulted in detrimental effects on fetal growth and infant mortality. However, in both cases the exposure was qualitatively different from that of chronic exposure from diet. In Basra, grain stores were treated with a fungicide containing organic mercury species, which led to mass poisoning in the following months [57]. In Minamata, an industrial plant contaminated the bay and marine life within it with organic mercury compounds [58]. In contrast, chronic dietary exposure as studied in this review frequently involves bio-methylation of elemental or inorganic species into methylmercury. Secondly, both contamination events involved far higher levels of mercury accumulation than any study in this review reported: hair mercury concentrations peaked in Basra at 120 to 600 μg/g [59] and in Minamata a maximum of 705 μg/g was reported [60]. It may be that the harms seen from acute poisoning are not generalizable to chronic exposure.”

3.6. Pg 12: “…and areas near industrial mercury use (e.g., [31,43])”. This sentence needs attention/clarification. The “industrial Hg use” refers to artisanal small-scale gold mining (ASGM) activities and loosely refers to fish in certain areas to have high-Hg concentrations for the case of Surinam [31] and in the case of [43] no attribution of fish-Hg consumption is related to ASGM activities. Please rewrite the sentence more accurately after reading the cited references.

This sentence has been rewritten to more accurately represent the range of pathways identified by study authors.

Two of these studies were conducted in Suriname and the Amazon basin, where the authors speculated that Hg exposure may have occurred respectively from fresh water fishing near small scale gold mining [26] and tin mining activities [38]. Studies in the Seychelles and coastal China reported that the expected route of exposure was diets high in ocean fish [33,44].

3.7. Table 2: Instead of “mercury source” use ‘sampled matrix’; mercury source conveys the idea of exposure pathways which should be explored. Please fix “exposure” in Appendix A.

Changed to “sampled matrix” as suggested.

3.8. Reference 39: Hong (2017) is a Doctoral dissertation referenced without year of publication. Because it is not a peer reviewed article (like the other studies) it should be justified or better, deleted.

Added a justification for why this and other non-peer reviewed articles were allowed to be included. We changed the layout of tables so that the paper can more easily be seen as different from published studies.

“We additionally used Google search to identify unpublished literature such as white papers, PhD dissertations or theses, or conference proceedings, where the full paper was digitally available. Including unpublished literature has the advantage of identifying studies where publication bias is less likely. Grey literature was subject to the same inclusion criteria and quality assessment as published studies.”

Round 2

Reviewer 3 Report

Authors appropriately addressed raised concerns.